# A Surrogate Model Based on Artificial Neural Network for RF Radiation Modelling with High-Dimensional Data

**DOI:** 10.3390/ijerph17072586

**Published:** 2020-04-09

**Authors:** Xi Cheng, Clément Henry, Francesco P. Andriulli, Christian Person, Joe Wiart

**Affiliations:** 1Chaire C2M, LTCI, Télécom Paris, 19 Place Marguerite Perey, 91120 Palaiseau, France; joe.wiart@telecom-paristech.fr; 2Department of Electronics and Telecommunications, Politecnico di Torino, IT-10129 Turin, Italy; clement.henry@polito.it (C.H.); francesco.andriulli@polito.it (F.P.A.); 3IMT Atlantique/Lab-STICC UMR CNRS 6285, Technopole Brest Iroise-CS83818-29238, 29238 Brest CEDEX 03, France; christian.person@imt-atlantique.fr

**Keywords:** artificial neural networks, uncertainty quantification, specific absorption rate

## Abstract

This paper focuses on quantifying the uncertainty in the specific absorption rate values of the brain induced by the uncertain positions of the electroencephalography electrodes placed on the patient’s scalp. To avoid running a large number of simulations, an artificial neural network architecture for uncertainty quantification involving high-dimensional data is proposed in this paper. The proposed method is demonstrated to be an attractive alternative to conventional uncertainty quantification methods because of its considerable advantage in the computational expense and speed.

## 1. Introduction

Good knowledge of the effect of wireless devices such as mobile phones and laptops on their surrounding environment is necessary in both the research community and the society at large. More precisely, it is essential to investigate the impact of electromagnetic waves on biological tissues [1]. In this scenario, a field of key importance is the study of the impact of electromagnetic radiation on brain activity. However, currently, none of the existing techniques allow the use of high resolution electroencephalography (EEG) devices without encountering a substantial shielding and field deformation which invariably oblige for high-resolution EEG recordings that are not simultaneous with the radio frequency (RF) radiation sources.

The final goal of our project is to design a system enabling high-resolution EEG recordings in the presence of an RF radiating source and taking electromagnetic field deformation into account. Achieving this goal requires understanding well the physical properties of the complex system such as the interaction between the metallic part of the EEG recordings and the RF source. Numerical modeling is an effective approach to investigate the properties of the new system instead of measurements which are expensive and time-consuming. The modeling of the radiating source, head, and EEG device relies on sets of input parameters which can affect the electromagnetic field, and thereby affect the specific absorption rate (SAR) values in the brain which is a measure of the rate at which energy is absorbed by the brain when exposed to an RF electromagnetic field [2]. In practice, the exact values of the inputs cannot be found, which produces uncertainties in the simulation results [3]. Uncertainty quantification (UQ) is indispensable when the acceptability of the simulation results is considered [3,4,5,6,7]. In this paper, UQ is focused on the analysis of uncertainties in the positions of the electrodes along the scalp. The modeling of uncertain positions of the electrodes and an artificial neural network (ANN) model for UQ are presented.

The traditional UQ approach is the Monte Carlo simulation (MCS) which requires running the deterministic code a large number of times to give accurate statistics, resulting in a high computational cost [3]. To solve the problem, surrogate models, which can be computed very efficiently, are often constructed to imitate the physical system concerned. This paper proposes a surrogate model which combines two different artificial neural networks (ANNs) [8,9] for UQ in the new EEG system modeling. ANNs are brain-inspired systems which aim to imitate how humans learn. Various advanced neural network structures have been investigated for a simple input–output relationship [9]. However, a simple input-output ANN has difficulty in handling a relatively small number of high-dimensional training samples since the insufficient training samples lead to inaccurate results due to the lack of training feature [10]. Therefore, effective feature learning methods are critically needed to automatically capture the useful features of the high-dimensional data such as SAR values observed in the brain. For the reasons mentioned above, an autoencoder neural network [8] is introduced for dimension reduction to obtain the features of the SAR values in the brain. The proposed ANN structure can be used to predict the corresponding features of SAR values for a new set of inputs. Then, the predicted SAR values in the brain corresponding to the new sets of inputs are reconstructed from the predicted feature by the decoder neural network. The statistical quantities associated with the SAR values in the brain such as the mean and standard deviation can be evaluated by running the proposed ANN model.

In summary, UQ is performed with the SAR values predicted by the proposed surrogate model. This paper is organized as follows. The proposed method is introduced in Section 2. Section 3 gives the description of the numerical simulation and the UQ results. Conclusion and perspectives are given in Section 4.

## 2. Proposed Surrogate Model for Uncertainty Quantification in RF Radiation Modeling

### 2.1. Numerical Modeling of the New EEG System

This section briefly describes the numerical solver used to obtain the SAR in a head covered by an EEG net in the presence of an RF source. The requirements for this electromagnetic field solver are to model both the metallic EEG caps and the head, which can be considered as perfectly electric conducting (PEC) surfaces and as an inhomogeneous lossy dielectric material, respectively. The solver chosen is based on the volume–surface integral equation (VSIE) [11,12]. It combines a surface integral equation (SIE) to model the PEC objects and a volume integral equation (VIE) to model inhomogeneous bodies. The couplings between the metallic and dielectric objects are included in the VSIE.

Consider a composite scatterer made of a PEC object with boundary Γ (can be either open or closed) and a linear inhomogeneous dielectric object Ω illuminated by a time-harmonic incident electromagnetic wave $E^{inc}$ in a background medium with permittivity $\epsilon_{0}$ and permeability $\mu_{0}$, as shown in Figure 1.

Ω has a complex permittivity $\epsilon_{c}\left(r\right)=\epsilon \left(r\right)-i\sigma \left(r\right)/\omega$ where ϵ(r) is the dielectric permittivity at position r∈Ω, σ(r) is the conductivity, and ω is the angular frequency. Using both the volume equivalence principle and the surface equivalence principle [13], we replace the dielectric bodies by an equivalent volume current density $J_{v}$ and the conducting objects by an equivalent surface current density $J_{s}$ defined on their surface. The volume current density is defined as
(1)$$J_{v}\left(r\right)=i\omega \kappa \left(r\right)D_{v}\left(r\right),$$
where *i* is the imaginary unit, $\kappa \left(r\right)=\left(\epsilon_{c}\left(r\right)-\epsilon_{0}\right)/\epsilon_{c}\left(r\right)$ is the dielectric contrast, and $D_{v}\left(r\right)$ is the electric flux.

The total field *E* can be written as a sum of the incident field and the scattered fields
(2)$$E\left(r\right)=E^{inc}\left(r\right)+E_{v}^{scatt}\left(r\right)+E_{s}^{scatt}\left(r\right),$$
where $E_{s}^{scatt}$ is the field scattered by $J_{s}$ and $E_{v}^{scatt}$ is the field scattered by $J_{v}$
(3)$$E_{v}^{scatt}\left(r\right)=\frac{k_{0}^{2}}{\epsilon_{0}}\int_{\Omega }G\left(r,r^{{}^{\prime }}\right)\kappa \left(r^{{}^{\prime }}\right)D_{v}\left(r^{{}^{\prime }}\right)dv^{{}^{\prime }}+\frac{1}{\epsilon_{0}}▽\int_{\Omega }G\left(r,r^{{}^{\prime }}\right){▽}^{{}^{\prime }}.\left(\kappa \left(r^{{}^{\prime }}\right)D_{v}\left(r^{{}^{\prime }}\right)\right)dv^{{}^{\prime }}$$
(4)$$E_{s}^{scatt}\left(r\right)=ik_{0}\eta_{0}\int_{\Gamma }G\left(r,r^{{}^{\prime }}\right)J_{s}\left(r^{{}^{\prime }}\right)ds^{{}^{\prime }}-\frac{\eta_{0}}{ik_{0}}▽\int_{\Gamma }G\left(r,r^{{}^{\prime }}\right){▽}^{{}^{\prime }}.J_{s}\left(r^{{}^{\prime }}\right)ds^{{}^{\prime }}$$
with *G* being the 3D Green’s function in vacuum and the constants $\eta_{0}$ and $k_{0}$ being the impedance and the wavenumber in vacuum, respectively.

The volume integral equation in Ω can be expressed as
(5)$$\begin{array}{c} \frac{D_{v}\left(r\right)}{\epsilon (r)}=E^{inc}\left(r\right)+E_{v}^{scatt}\left(D_{v}\left(r\right)\right)+E_{s}^{scatt}\left(J_{s}\left(r\right)\right)\;\forall r\in \Omega . \end{array}$$

On a PEC object, the boundary condition requires that the tangential component of the total electric field vanishes. This gives the surface integral equation on Γ
(6)$$\begin{array}{c} \hat{n_{r}}\times E^{inc}\left(r\right)=\hat{n_{r}}\times E_{v}^{scatt}\left(D_{v}\left(r\right)\right)+\hat{n_{r}}\times E_{s}^{scatt}\left(J_{s}\left(r\right)\right)\;\forall r\in \Gamma , \end{array}$$
where $\hat{n_{r}}$ is the surface normal vector.

The VSIE is defined by Equations (5) and (6) and is solved for $J_{s}$ and $D_{v}$. The next step is to discretize those equations into a matrix system using the method of moments (MoM) [13]. The volume Ω is discretized with tetrahedra and the surface Γ with triangular patches. Note that the triangular patches must coincide with the faces of the tetrahedra at the junctions between Ω and Γ. Rao–Wilton–Glisson (RWG) basis functions $f_{m}^{s}\left(r\right)$ [14] and Schaubert–Wilton–Glisson (SWG) basis functions $f_{m}^{v}\left(r\right)$ [15] are used to discretize the unknowns $J_{s}$ and $D_{v}$, respectively
(7)$$\begin{array}{c} D_{v}\left(r\right)=\sum_{m=1}^{M_{v}}\alpha_{m}f_{m}^{v}\left(r\right) \end{array}$$
(8)$$\begin{array}{c} J_{s}\left(r\right)=\sum_{m=1}^{M_{s}}\beta_{m}f_{m}^{s}\left(r\right), \end{array}$$
where $M_{v}$ is the number of SWG functions and $M_{s}$ is the number of RWG basis functions.

Applying this discretization and testing the equation with both SWG and RWG basis functions, we obtain a block matrix system
(9)$$\begin{array}{c} \left[\begin{array}{cc} Z_{vv} & Z_{sv}\\ Z_{vs} & Z_{ss} \end{array}\right]\left[\begin{array}{c} \alpha \\ \beta \end{array}\right]=\left[\begin{array}{c} v_{v}\\ v_{s} \end{array}\right] \end{array}$$

In Equation (Equation 9), the diagonal blocks are respectively the standard VIE and SIE and the off-diagonal block represent the coupling between the volume and surface scatterers. The excitation vector is $v_{v}$ for the volume part and $v_{s}$ for the surface part. The system is solved for the unknown expansion coefficients α and β.

The total electric field *E* in Ω is directly related to $D_{v}$ and the SAR within a tissue of the head can be obtained from the average norm of *E* in that tissue
(10)$$SAR=\frac{\sigma {\left|E\right|}^{2}}{2\rho },$$
where ρ is the tissue density and σ is its conductivity.

### 2.2. Design of Experiments (DoE)

The head model is a three-layer sphere discretized with tetrahedra. The PEC electrodes are formed by the exterior triangles of the tetrahedra pertaining to the boundary of the discretized sphere. The discretized geometry obtained is shown in Figure 2. In this case, the position of each electrode can be represented by $\left(r,\theta_{p},\phi_{p}\right)$ (p=1,…,L) in a spherical coordinate system, where *r* is the radius of the sphere, $\theta_{p}$ is the polar angle, $\phi_{p}$ is the azimuthal angle, and *L* is the number of electrodes. The original Cartesian coordinates of the electrodes $\left(P_{p}^{x},P_{p}^{y},P_{p}^{z}\right)$ are obtained from a toolbox, and are used in the numerical simulations. In the following section, the uncertainties in the positions of the electrodes are modeled in the spherical coordinate system, and the coordinates with uncertainties are required to be transformed into the Cartesian coordinates for numerical simulation.

In the spherical coordinate system, the center of each electrode is moved in a square, and the size of the square is controlled by ∆. To simplify the problem, 9 possible positions of the center for each electrode are chosen and represented by 9 indices, which are shown in Figure 3. The relationship between the indices and the possible positions of the center is provided in Table 1. When ∆ is determined, the uncertain position of an electrode can be modeled by the nine indices, which makes it a discrete random variable. In the above-mentioned case, the electrodes’ positions are changed independently of one another, and the number of combinations is $9^{L}$.

There are three specific issues with this modeling. The first one is that the total CPU time for a single SAR simulation is about 25 h when performed on a computer cluster with 32 cores, which makes impossible the use of MCS directly with the solver. Second, the distance moved by the center of each electrode must be greater than the edge length of the triangle otherwise the algorithm used in the toolbox will arrange the electrode in its original position. This means there is a discrete variation of the uncertain inputs and this variation must be relatively large. Third, the SAR values in brain are high-dimensional data, and there is a small number of high-dimensional SAR values available. To solve the aforementioned problems, an ANN model for UQ is presented in the following section.

### 2.3. Proposed Surrogate Model for UQ

The structure of the new surrogate model is shown in Figure 4. Since the output (SAR values observed in the brain) of the numerical simulation is high-dimensional, it is difficult to handle it with a conventional input-output structure of ANN. Therefore, a pre-trained autoencoder neural network is introduced into the proposed surrogate model for dimensionality reduction to map the high-dimensional outputs to a suitable low-dimensional space, and also for reconstructing the original high-dimensional data.

First of all, an autoencoder neural network is trained. It can be divided into two separate networks: an encoder and a decoder. The structure of an autoencoder neural network is presented in Figure 5. Given the training data $\mathrm{SAR}=\{{\mathrm{SAR}}_{1},{\mathrm{SAR}}_{2},{\mathrm{SAR}}_{3},\ldots ,{\mathrm{SAR}}_{N}\}$ (${\mathrm{SAR}}_{n}$ (1≤n≤N) represents a D-dimensional vector ${\mathrm{SAR}}_{n}\in R^{D}$), the encoder transforms the input matrix SAR into a hidden representation $C=\{C_{1},C_{2},C_{3},\ldots ,C_{N}\}$ ($C_{n}$ represents a d-dimensional vector $C_{n}\in R^{d}$) through activation functions, where d≪D, and *N* is the number of input samples for the autoencoder neural network. Then, the matrix C is transformed back to a reconstruction matrix ${\mathrm{SAR}}^{\prime }=\left\{{\mathrm{SAR}}_{1}^{\prime },{\mathrm{SAR}}_{2}^{\prime },{\mathrm{SAR}}_{3}^{\prime },\ldots ,{\mathrm{SAR}}_{N}^{\prime }\right\}$ (${\mathrm{SAR}}_{n}^{\prime }$ is a D-dimensional vector ${\mathrm{SAR}}_{n}^{\prime }\in R^{D}$) by the decoder.

Subsequently, the proposed ANN is trained. The input samples of the proposed ANN are $I=\{I_{1},I_{2},I_{3},\ldots ,I_{M}\}$ ($I_{m}$ (1≤m≤M) represents a S-dimensional vector $I_{m}\in R^{S}$), which are the uncertain inputs of the EEG numerical simulation, and $C^{\prime }=\left\{C_{1}^{\prime },C_{2}^{\prime },C_{3}^{\prime },\ldots ,C_{M}^{\prime }\right\}$ ($C_{m}^{\prime }$ represents a d-dimensional vector $C_{m}^{\prime }\in R^{d}$), which are the predicted features of SAR from the encoder neural network, where *M* is the number of input samples for the proposed ANN.

Finally, in the testing process, the compressed codes $C^{\prime }$ can be obtained from the proposed ANN for a new set of uncertain inputs I, and then using the decoder, the predicted outputs $U^{\prime }$ corresponding to the new set of inputs I are obtained. The proposed ANN model can predict the outputs of the numerical simulation very quickly. The statistical quantities of SAR values in brain can be evaluated by running the surrogate model instead of running numerous of numerical simulations.

The hyperparameters of the two ANNs such as the number of hidden layers and units, activation function, learning rate, etc., depending on the data of the problem. In this work, the rectified linear unit (Relu) function [16] is used as the activation function in both the hidden layers of the autoencoder neural network and the proposed ANN. The Relu function is
(11)$$f\left(a\right)=\left\{\begin{array}{ccc} 0 & for & a<0\\ a & for & a\geq 0 \end{array}\right.$$
and the linear activation function [17] is used as activation function in the input layer and the output layer of the ANNs. The linear activation function is
(12)f(a)=a
where *a* is the input to a neuron. The backpropagation algorithm is used to train the two neural networks, and the parameters are optimized through adaptive moment estimation (Adam) [17]. The mean squared error (MSE) is used for performance evaluation in the ANN
(13)$$MSE=\frac{1}{R}\sum_{r=1}^{R}{\left(Y_{r}-\hat{Y_{r}}\right)}^{2}$$
where $Y_{r}$ and $\hat{Y_{r}}$ denote the observed and forecasted values, respectively, of the *r*th datum, and *R* is the total number of the data.

## 3. Simulations and Results

### 3.1. Model Description

As briefly described in Section 2.2, the model used for the head is the three-layer sphere presented in Figure 2b. The parameters used to define the model are listed in Table 2 for a frequency of 900MHz. The 64 EEG caps are defined as hexagons on the surface of the sphere as shown in Figure 2a. The mean diameter of an electrode cap is 13mm. The SAR simulations are performed on a computer cluster with 32 cores (2× Intel Xeon Scalable Processors Gold 6130 2.10 GHz 16 cores). The total CPU time for a single SAR simulation is 90,327 seconds (s) (about 25 h).

### 3.2. Results and Discussion

The SAR values in the brain obtained from a single simulation are shown in Figure 6. They are transformed into a vector to be used by the proposed ANN model as shown in Figure 7. The ratio of the standard deviation to the mean of the SAR values is obtained for 500 different configurations of the EEG caps and presented in Figure 8. It shows a substantial variation of the SAR values in the brain induced by the uncertain positions of the EEG caps. The indices of the electrodes, and the SAR values in brain are both normalized before being inserted into the surrogate model. The dimensions of the one-dimensional SAR values in the brain and its codes are 27,000 and 100, respectively. There are 100 training samples and 400 testing samples for the autoencoder neural network. The proposed neural network has 200 training samples and 200 testing samples. To perform a validation, the training samples of each ANN are split into two parts: a training set and a validation set. The validation data represent 30% of the training data. The average training, validation, and testing MSE values from the autoencoder neural network and the proposed ANN are given in Table 3. The hyperparameters of the ANN model are presented in Table 4. The mean and standard deviation of the SAR values in the brain are predicted from the proposed surrogate model and presented in Figure 9. The prediction results of the proposed method are compared with those of full-wave simulations, and they have a good consistency. The CPU time required for the new method includes two parts: (1) the time to train the autoencoder neural network and the proposed ANN, and (2) the time to test the proposed ANN structure. The detail of the CPU time of the UQ method is presented in Table 5. It shows that when the ANN model is trained, it can predict the outputs of the full-wave simulations quickly.

Compared with running plenty of full-wave simulations in MCS, the proposed sorrogate model largely reduces the number of simulations and improves the efficiency. In this paper, all calculations of the ANN model are performed on an Intel i7-6500U 2.6 GHz machine with 8 GB of RAM.

## 4. Conclusions

The work presented aims to quantify the uncertainty of SAR values in the brain induced by the uncertain positions of EEG electrodes on the scalp when the acceptability of the simulation result is considered. MCS is intractable in this scenario since the simulation time would be too long. The proposed surrogate model can perform UQ with high-dimensional data and highly varying inputs. In addition, it does not require prior knowledge of the uncertain input parameters of a numerical simulation, such as their probability distributions. The prediction results of the proposed surrogate model show a good agreement with the results of the full-wave simulations, and the proposed method is superior to MCS in consideration of the computational expense and speed. Future work will focus on further reducing the number of training samples required for the surrogate model.

## Figures and Tables

**Figure 1 ijerph-17-02586-f001:**
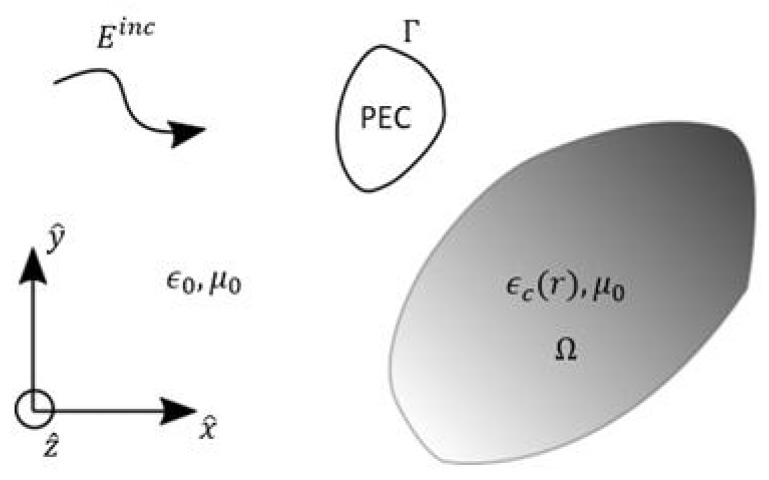
Description of the 3D scattering problem. The perfectly electric conducting (PEC) parts are modeled by their boundary (i.e., Γ) and the dielectric parts are modeled by their volume (i.e., Ω).

**Figure 2 ijerph-17-02586-f002:**
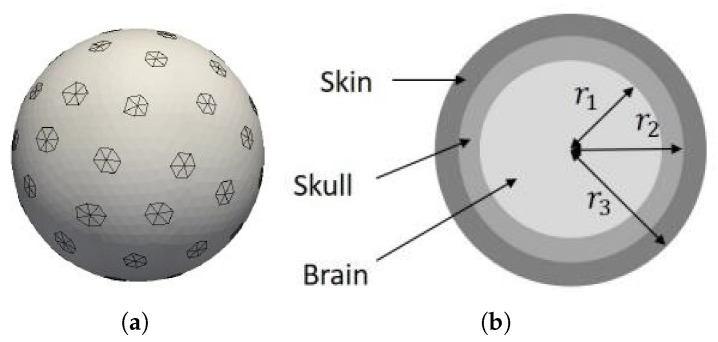
Head model with electroencephalography (EEG) electrodes: (**a**) PEC electrodes formed by triangles lying on the surface of the discretized three-layer sphere, and (**b**) three-layer spherical head model.

**Figure 3 ijerph-17-02586-f003:**
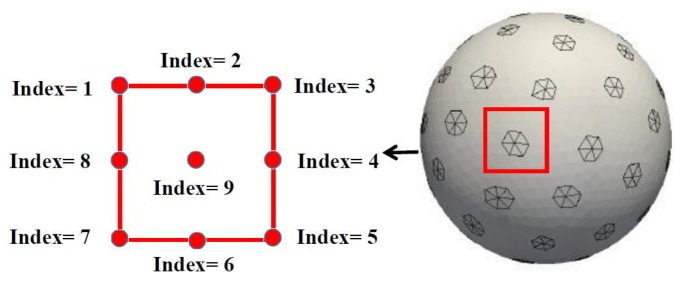
Indexes of the nine possible positions of an electrode.

**Figure 4 ijerph-17-02586-f004:**
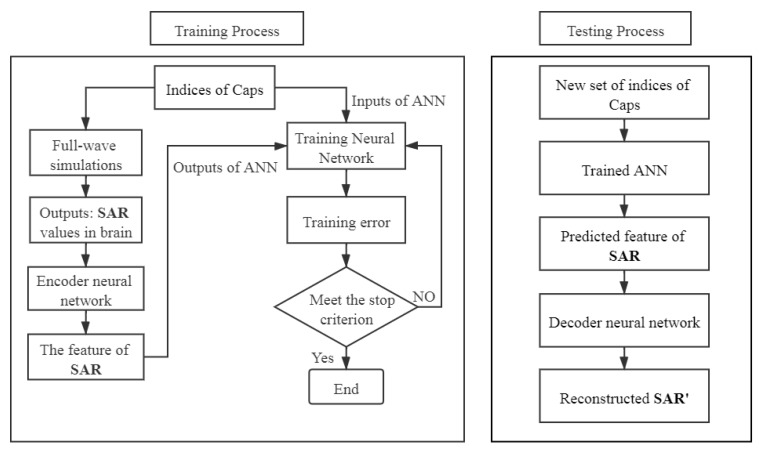
Structure of the proposed surrogate model for uncertainty quantification (UQ).

**Figure 5 ijerph-17-02586-f005:**
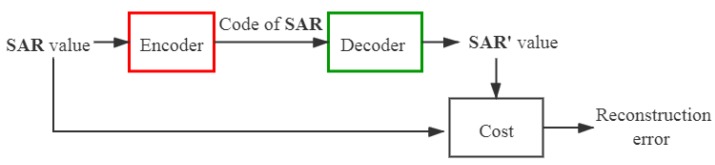
Structure of an autoencoder neural network.

**Figure 6 ijerph-17-02586-f006:**
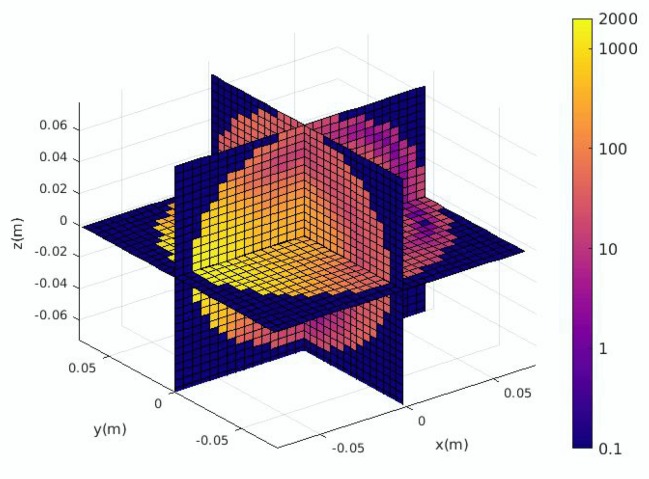
SAR values in brain (W/kg).

**Figure 7 ijerph-17-02586-f007:**
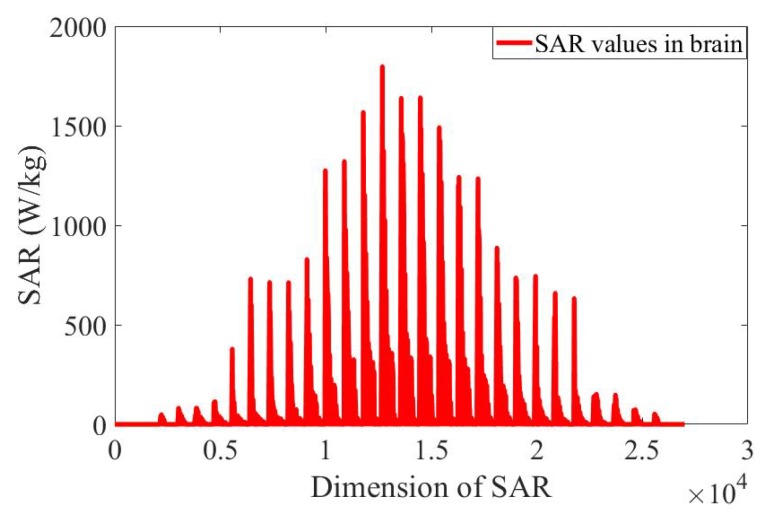
Specific absorption rate (SAR) values in brain transformed into one-dimensional data.

**Figure 8 ijerph-17-02586-f008:**
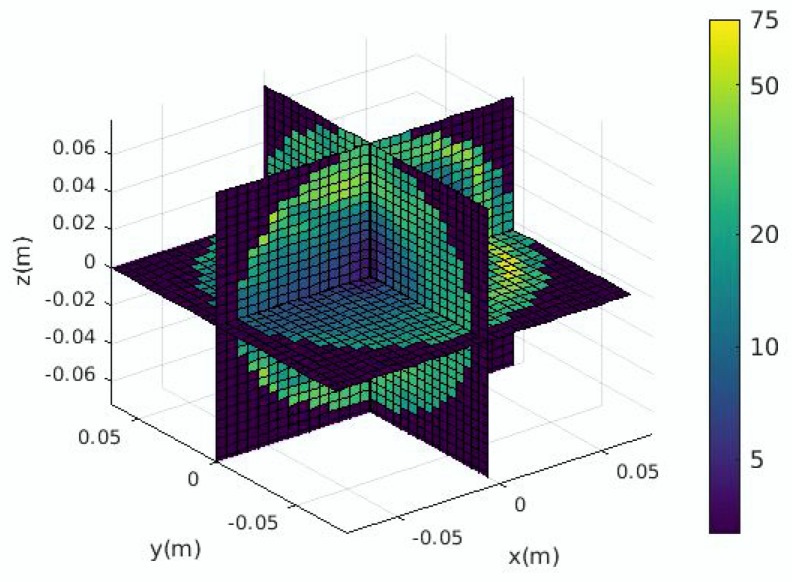
Ratio of the standard deviation to the mean of the SAR values in brain (%).

**Figure 9 ijerph-17-02586-f009:**
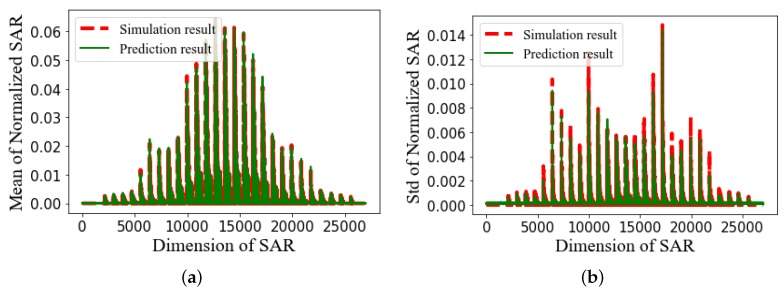
(**a**) Mean of the normalized SAR in brain, and (**b**) standard deviation of the normalized SAR in brain.

**Table 1 ijerph-17-02586-t001:** Relationship between the index and the position.

Index 1	($\theta_{p}-∆,\phi_{p}-∆$)
Index 2	($\theta_{p}-∆,\phi_{p}$)
Index 3	($\theta_{p}-∆,\phi_{p}+∆$)
Index 4	($\theta_{p},\phi_{p}+∆$)
Index 5	($\theta_{p}+∆,\phi_{p}+∆$)
Index 6	($\theta_{p}+∆,\phi_{p}$)
Index 7	($\theta_{p}+∆,\phi_{p}-∆$)
Index 8	($\theta_{p},\phi_{p}-∆$)
Index 9	($\theta_{p},\phi_{p}$)

**Table 2 ijerph-17-02586-t002:** Parameters used for the three-layer sphere at 900 MHz (four-year child).

	Relative Permittivity	Conductivity (S/m)	Mass Density (kg/m${}^{3}$)	Outer Radius (mm)
Brain	55.5	0.94	1030	68
Skull	12.5	0.14	1850	71.1
Skin	35.2	0.6	1110	74.7

**Table 3 ijerph-17-02586-t003:** Training error, validation error, and testing error of the surrogate model.

Network	Training		Testing
	Training Error	Validation Error	Testing Error
Autoencoder	1.29 × $10^{-7}$	1.68 × $10^{-7}$	1.74 × $10^{-7}$
Proposed ANN	4.92 × $10^{-7}$	5.16 × $10^{-7}$	5.45 × $10^{-7}$

**Table 4 ijerph-17-02586-t004:** Hyperparameters of the surrogate model.

Network	Number of Training Data	Batch Size	Number of Epochs	Number of Neurons in Each Layer
Autoencoder	100	25	8000	3000, 3000, 3000, 100, 3000, 3000, 3000
Proposed ANN	200	50	6000	500, 500, 100, 100

**Table 5 ijerph-17-02586-t005:** CPU time to train and test the proposed surrogate model.

	Number of Data	CPU Time
Training	100 + 200	172,036 s + 291 s
Testing	200	2 s

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
