# Peer review of "A Surrogate Model Based on Artificial Neural Network for RF Radiation Modelling with High-Dimensional Data"

_ijerph, 2020, doi:10.3390/ijerph17072586_

Round 1

Reviewer 1 Report

The manuscript is well written and interesting. However, the current manuscript lacks of details in particular on the construction of the surrogate model. 

More results from the training and validated data are needed. It is unclear how the ANN performs since only the performance has not been quantified, neither provided. 

The paper is about UQ but no results about UQ are provided. Either the title of the paper is changed or results of UQ are provided. 

The authors in calculating the mean and std of the output assumed that all the configuration are equi probable. A very strong assumption.  

Additional comments are as follows:

Line 14: What is the meaning of EEG?

Line 26: Why are random input? The may be unknown, not necessary random. 

Line 27: It is more related to the credibility of the model (aka model validation and verification) (see e.g. https://royalsocietypublishing.org/doi/full/10.1098/rsos.180687 or https://www.sciencedirect.com/science/article/pii/S0045782507005105 orhttps://www.cambridge.org/core/books/verification-and-validation-in-scientific-computing/05CA1F8F3CCB5AE5445FDF55239A0183)

Line 31: don't use MCM please. Either MC or MCS (Monte Carlo Simulation)

Line 35: why mentioning PCE if not used? It seems the authors need to justify  for not having used PCE. But PCE is only one of the surrogate model there. 

Line 60: "UQ is performed with a relatively small number of numerical simulations by the proposed surrogate model." Why is this important or needed. Once the ANN is available any type of analysis can be performed. 

Line 107: Please specify the computational cost of the numerical simulation of EEG (although mentioned at line 154).

Line 111: we all know that every single method has limitation. No needed to justify the non selection of PCE.

Line 144: The authors might also want to consider the error introduced by the ANN or the uncertainty associated to the training of such surrogate model (see e.g. https://doi.org/10.1016/j.neunet.2017.09.003 or https://doi.org/10.1109/SSCI.2017.8285163  )

Section 3.1 

Are those parameters assumed to be known perfectly?

Section 3.2

It is not clear why the representazion of the output in a single vector has been used. Is there a real advantage to have a single output? ANN are really good mapping images expecially using convolution neural network. Therefore it is not clear the real advantage of this modellisation.

In conclusion, the manuscript requires significant improvments.  

Reviewer 2 Report

 The compucational expense and speed must be given for the present method

and the traditional UQ in this paper。

Round 2

Reviewer 1 Report

The authors have provided a much improved version of the manuscript. All the major criticisms have been addressed properly. 

Therefore, the manuscript may be consider for a publication in this journal.